|---------|--------|-----------|---------------------------|----------------|
| Validation Label | Grad Dot | 0.084 | 0.483 | 0.999 |
| | Grad Sim | 0.106 | 0.647 | 1.000 |
| | LESS | 0.388 | 0.724 | 1.000 |
| | DataInf | 0.204 | 0.487 | 0.999 |
| | EKFAC | 0.216 | 0.497 | 1.000 |
| Model Response | Grad Dot | 0.070 | 0.006 | 0.004 |
| | Grad Sim | 0.097 | 0.006 | 0.003 |
| | LESS | 0.097 | 0.006 | 0.002 |
| | DataInf | 0.100 | 0.006 | 0.004 |
| | EKFAC | 0.100 | 0.006 | 0.007 |

setting). A significant drop in performance is observed when using unsafe model responses (responses that receive an maximum unsafe scoring of 5), which results in poor detection accuracy. As noted in prior work [40], relying on model responses can introduce noisy gradient features, thereby failing to provide a reliable signal for identifying unsafe training data.

Moreover, for benchmarking purposes, using validation labels provides a consistent and stable set of targets across all methods, enabling fair and reliable comparisons. In contrast, using model responses introduces variability, as different models may produce different outputs for the same input. This inconsistency can lead to instability in attribution signals and compromise the fairness and reproducibility of evaluation.

## D.9 Retrain model safety

Table 6: Retrain model safety after removing top **30** identified data (*Homogenous* setting). A higher GPT score/ASR indicates a more toxic/biased model (See Appendix D.3). Results average over 5 inference runs.

| Model | Filtering Method | ToxicChat GPT score | ToxicChat GPT ASR | XSTest-response GPT score | XSTest-response GPT ASR | JailbreakBench GPT score | JailbreakBench GPT ASR |
|-------|-----------------|-----------|---------|-----------|---------|-----------|---------|
| Pythia-1b | Rep Sim | 2.04 | 8.7 | 4.14 | 60.0 | 3.73 | 45.3 |
| | GradSafe | 1.86 | 7.3 | 4.22 | 64.0 | 3.76 | 44.7 |
| | Wildguard | 2.11 | 7.3 | 4.44 | 80.0 | 3.69 | 48.0 |
| | LESS | 2.01 | 5.3 | 4.02 | 58.0 | 3.69 | 42.0 |
| Llama3.2-1B | Rep Sim | 2.18 | 12.7 | 4.08 | 60.0 | 3.94 | 54.0 |
| | GradSafe | 2.14 | 14.0 | 3.96 | 62.0 | 3.75 | 48.0 |
| | Wildguard | 2.25 | 17.3 | 4.14 | 66.0 | 3.93 | 49.3 |
| | LESS | 2.01 | 10.0 | 3.80 | 54.0 | 3.77 | 47.3 |
| Llama3.1-8B | Rep Sim | 2.87 | 29.3 | 4.44 | 80.0 | 4.42 | 74.0 |
| | GradSafe | 2.90 | 30.7 | 4.62 | 80.0 | 4.21 | 70.0 |
| | Wildguard | 2.79 | 27.3 | 4.70 | 86.0 | 3.79 | 58.7 |
| | LESS | 2.8 | 22.7 | 4.30 | 72.0 | 3.63 | 55.3 |

Detection of unsafe training data is crucial for curating safer datasets and improving model safety through retraining. To further evaluate the filtering abilities of different detection methods, we conduct retraining experiments where the top **30** unsafe training examples—identified by baseline methods and data attribution methods—are removed. For data attribution, we use the **LESS** method, which demonstrates strong and consistent performance across most experiments.