# OpenReview forum: "DATE-LM: Benchmarking Data Attribution Evaluation for Large Language Models"
_NeurIPS.cc/2025/Datasets_and_Benchmarks_Track — NeurIPS 2025 Datasets and Benchmarks Track poster_

### Official Review · Reviewer_uP4U · 2025-06-30

**Rating:** 4
**Confidence:** 3

**Summary:**

This paper establishes a data attribution framework based on large language models and analyzes its performance and accuracy across three scenarios and eight attribution/non-attribution algorithms. However, the practical applicability of data attribution methods still requires further investigation.

**Dataset Code Accessibility:**

Yes

**Ethical Considerations:**

No, there are no or only very minor ethics concerns

**Final Justification:**

The author's response addresses my concerns.

**Limitations Weaknesses:**

- According to the results in Table 2, the improvements achieved by various data selection methods over random selection are minimal. For example, at 10k steps, the best-performing baseline improves by only 0.66%, while incurring 4.3$\times$ the cost--some methods even increase the cost by up to 11$\times$. Similar patterns are observed at 30k steps, where advanced data selection methods offer marginal gains at significant computational cost. Thus, it is difficult to justify the practical value of data selection in pre-training tasks.
- The results in Figure 2 further show that all data selection methods are highly sensitive to the temperature coefficient hyper-parameter. Under the high-cost setting of data selection, this sensitivity further limits their practical applicability.
- In Table 3, most non-attribution-based data selection methods are computationally expensive, and Figure 3 shows that attribution-based methods are highly sensitive to stylistic similarity. These observations further undermine the practical significance of data selection.
- Although this is a benchmark-oriented work, the usefulness of the data attribution field itself will also impact the contribution of this paper to the community.
- One of the key conclusions of the paper is that "no single attribution method dominates the others," which raises practical concerns about how to select the appropriate method in real-world applications. The authors should provide a summarized guideline or recommendation to help readers choose suitable methods in practice.
- The data attribution process requires a reference dataset $D_{\text{ref}}$. In certain specific or low-resource domains/tasks, how to obtain or construct such a dataset remains an open question.
- The leaderboard link on page 2 is broken and should be corrected.

**Strengths Contributions:**

- This paper establishes a data attribution framework and evaluates its practical applications across various scenarios, including pre-training/fine-tuning, bias filtering, and factual attribution.
- This paper provides performance and efficiency analysis of both non-attribution and attribution-based methods under different applications.
- The paper is well-structured and clearly organized.

---

> ### Author Rebuttal · Authors · 2025-07-31
>
> We thank the reviewer for taking the time to provide us their comments, and we're glad that the reviewer found our paper to be well-structured and organized. We will address each of your questions/concerns below:
>
> ---
>
> **According to the results in Table 2, the improvements achieved by various data selection methods over random selection are minimal ... Thus, it is difficult to justify the practical value of data selection in pre-training tasks.**
>
> We thank the reviewer for raising this important point. We highlight that the goal of DATE-LM is not to propose new data selection methods, but to assess the practical utility of attribution methods across multiple LLM applications, and allow researchers to identify gaps in current methods in order to improve tradeoffs between computation costs and performance.
>
> Indeed, one of the findings of our benchmark is that some data selection methods offer limited gains over random selection in pre-training settings -- this is not a shortcoming, but rather aligns with our benchmark’s motivation. We further emphasize that we include the pre-training setting because it is gaining significant attention in the community, as seen in recent efforts like DataComp-LM [1] and MATES [2], making it important to comprehensively evaluate data selection methods in this realm.
>
> Additionally, in other settings such as factual attribution, data attribution methods in our benchmark consistently outperform non-attribution baselines, which highlights their practical value in other important realms. We believe that unifying and standardizing evaluation across such tasks, our framework will help the community identify how data attribution methods are most effective for LLMs.
>
> [1] Li et al. DataComp-LM: In search of the next generation of training sets for language models. 2024
>
> [2] Yu et al. MATES: Model-Aware Data Selection for Efficient Pretraining with Data Influence Models. Neurips 2024.
>
> ---
>
> **The results in Figure 2 further show that all data selection methods are highly sensitive to the temperature coefficient hyper-parameter. Under the high-cost setting of data selection, this sensitivity further limits their practical applicability.**
>
> We appreciate the reviewer’s observation that data selection methods are sensitive to the Gumbel-top-k temperature, as shown in Figure 2. We agree, and emphasize that this is also a central insight of DATE-LM rather than a shortcoming. Sensitivity to data selection hyperparameters is a common phenomenon. Despite this, many widely used methods combine quality and diversity mechanisms, and have proven effective in practice [1, 2, 10, 11] , underscoring the need for a fair and modular benchmark like DATE-LM to support systematic evaluation for existing and future methods.
>
> [10]  Wettig et al. QuRating: Selecting High‑Quality Data for Training Language Models. ICML 2024.
> [11] Penedo et al. The FineWeb Datasets: Decanting the Web for the Finest Text Data at Scale. Neurips 2024.
>
> ---
>
> **In Table 3, most non-attribution-based data selection methods are computationally expensive**
>
> Thank you for the observation. We would like to note that not all non-attribution-based data selection methods are computationally expensive. For instance, text-based non‑attribution baselines like BM25 run in near‑real‑time, making them orders of magnitude faster than attribution methods while still showing improvement compared to random (Table 2)  -- thus, our framework provides a platform for researchers to be able to thoroughly compare attribution/non-attribution methods.
>
> ---
>
> **Figure 3 shows that attribution-based methods are highly sensitive to stylistic similarity. These observations further undermine the practical significance of data selection.**
>
> We thank the reviewer for raising the issue of sensitivity in our data‐selection methods. We highlight that our benchmark is designed to assess stylistic sensitivity in order to provide robust evaluation of data attribution methods. From a practical perspective, we observe that despite these sensitivities, data‐attribution methods still outperform baselines across various settings — both on detection tasks (toxicity/bias and factual attribution; Tables  5 and  9) and in retraining experiments, where they have competitive downstream performance compared to model‑agnostic or data‑agnostic approaches (Tables 12 and 14).
>
> From a diagnostic perspective, our benchmark was able to provide interesting insights to improve on the stylistic sensitivity of attribution-based methods – particularly in toxicity/bias detection, when the injected unsafe data are diverse, inner‑product–based attribution can over‑emphasize surface similarity. Figure 3 exposes this sensitivity and pinpoints exactly where attribution struggles, offering insights for developing more robust and efficient future data attribution methods.
>
> ---
>
> **Although this is a benchmark-oriented work, the usefulness of the data attribution field itself will also impact the contribution of this paper to the community.**
>
> We appreciate the reviewer’s observation. We note that data attribution as a field is becoming increasingly relevant in LLM research, given that it offers a promising path toward LLM data curation, bias mitigation, and data valuation as already noted in several works [2, 3, 4, 5, 6, 7]. We further emphasize that our benchmark provides a platform for researchers to continue making progress in this area, for example our benchmark has already shown:
>
> 1. Proven solutions: In scenarios where the attribution signal is relatively clean—such as toxicity/bias filtering and factual‐attribution tasks—data attribution methods like Grad‑Sim and LESS consistently outperform all baselines (see Tables 5, 9, 12, 14). This demonstrates that attribution can be adopted by the community as an effective, off‑the‑shelf solution for various LLM tasks.
>
> 2. Identification of key gaps: We pinpoint settings where attribution fails (e.g., sensitivity to stylistic similarity and under‑coverage of rare examples) across toxicity/bias, factual, and training‑data selection tasks.
>
> To this end, our benchmark shows that there are advances that data attribution methods have already achieved, and lay the groundwork for their future development.
>
> [3] Xia et al. LESS: Selecting Influential Data for Targeted Instruction Tuning. ICML 2024.
>
> [4] Pan et al. Detecting and Filtering Unsafe Training Data via Data Attribution. 2025
>
> [5] Zhang et al., Fairshare Data Pricing via Data Valuation for Large Language Models. 2025
>
> [6] Grosse et al. Studying Large Language Model Generalization with Influence Functions. 2023.
>
> [7] DataInf: Efficiently Estimating Data Influence in LoRA-tuned LLMs and Diffusion Models. ICLR 2024.
>
> ---
>
> **One of the key conclusions of the paper is that "no single attribution method dominates the others," which raises practical concerns about how to select the appropriate method in real-world applications. **
>
> Thank you for this suggestion. We agree that selecting the appropriate attribution method is challenging in practice. However, the goal of DATE-LM is not to prescribe a "best" method, but rather to enable systematic comparison across real-world LLM applications, so that researchers/practitioners can make informed decisions. While the best method is context-specific, we give several practical insights in the paper:
>
> 1. In training data selection and toxicity bias, non-attribution baselines like RepSim currently match or outperform more complex attribution methods, especially when cost is factored in.
>
> 2. In factual attribution, data attribution methods (specifically Grad Sim and LESS) show top-ranking performance, particularly in challenging setups (e.g., heterogeneous distractors, counterfactual tracing) where text based similarity is insufficient, and have potential for future gains if stronger methods can be developed.
>
> Thus, while current attribution methods may not yet be ideal for all settings, DATE-LM provides the foundation for developing/evaluating future methods, which is important as data attribution continues to gain traction in LLM research. We will make these takeaways more explicit in our next version.
>
> ---
>
> **The data attribution process requires a reference dataset . In certain specific or low-resource domains/tasks, how to obtain or construct such a dataset remains an open question.**
>
> Thank you for noting this important point. While there are recent works which examine how to best construct a reference set [8, 9], especially in low-resource domains, in DATE-LM, we follow the standard practice established in prior data attribution work:
>
> In the training data selection task, we use randomly sampled reference data from a reserved portion of the training set, as typically done in previous works [2,3,4]. For other applications such as toxicity/bias filtering and factual attribution, we conducted the reference set post hoc based on model behavior: we first train a model on the full dataset, then collect its outputs and identify examples that exhibit the target behavior (e.g., toxicity or factual error). This reflects a practical scenario where reference examples may come from developer-curated or user-reported outputs, making this feasible in some low-resource settings.
>
> Note that if practitioners wish to research the construct and effectiveness of different reference sets, our evaluation framework allows them to easily plug in their reference set for comprehensive large scale evaluation. We will include the above discussion in the next version of our paper.
>
> [8] Li et al. One-Shot Learning as Instruction Data Prospector for Large Language Models. ACL 2024
>
> [9] Ilyas et al. Datamodels: Predicting Predictions from Training Data. 2022.
>
> ---
>
> **The leaderboard link on page 2 is broken and should be corrected.**
>
> Thank you for pointing this out. The correct link is: https://huggingface.co/spaces/DataAttributionEval/DATE-LM-Leaderboard. We will update this in our paper.

---

> > ### Comment · Reviewer_uP4U · 2025-08-05
> > **response to the authors**
> >
> > Thank you for the detailed response. My concerns have been addressed, and I will raise my rating.

---

### Official Review · Reviewer_AcxT · 2025-07-02

**Rating:** 5
**Confidence:** 3

**Summary:**

This paper presents DATE-LM, a unified benchmark for evaluating data attribution methods in large language models. DATE-LM evaluates attribution quality through three key LLM applications: training data selection, toxicity/bias filtering, and factual attribution. It offers a modular pipeline, pretrained checkpoints, and a public leaderboard. Benchmark results show that no method excels universally, and simple baselines can match complex approaches, underscoring the need for application-specific, robust evaluation.

**Dataset Code Accessibility:**

Yes

**Ethical Considerations:**

No, there are no or only very minor ethics concerns

**Final Justification:**

My concerns have been addressed.

**Limitations Weaknesses:**

1. There is insufficient introduction to existing benchmarks and how the proposed benchmark improves upon them. It is recommended to provide more detailed explanations in next version, preferably with a clear, itemized list for clarity.
2. The use of large language models (LLMs) for evaluation is not clearly explained. It is unclear whether the same LLM is used across the three application scenarios. Additionally, it is important to clarify how the accuracy of each data attribution method is ensured during the evaluation process.

**Strengths Contributions:**

1. The benchmark is application-oriented, encompassing training data selection, toxicity and bias filtering, and factual attribution.
2. It features a modular design and includes pre-trained models and public leaderboards, enabling fair comparisons across multiple dimensions.
3. Insightful findings highlighting method trade-offs and the limits of attribution under real-world conditions.

---

> ### Author Rebuttal · Authors · 2025-07-31
>
> We thank the reviewer for taking the time to provide us their comments. We will address each of your questions/concerns below:
>
> ---
>
> **There is insufficient introduction to existing benchmarks and how the proposed benchmark improves upon them. It is recommended to provide more detailed explanations in next version, preferably with a clear, itemized list for clarity.**
>
> We thank the reviewer for highlighting the need to situate DATE‑LM among prior benchmarks. Across all three evaluation tasks, DATE‑LM makes the following advances:
>
> - Training-Data Selection: DataComp-LM (DCLM) [1] is the most related benchmark; it is a pretraining data selection benchmark where users can submit any new dataset, or a filtered subset of DCLM-Pool. However, our benchmark uses standardized mechanisms to fairly compare scoring functions, and incorporates cost‑saving measures [Section 4.1]; We also support instruction-finetuning setting in addition to pretraining.
> - Toxicity/Bias Filtering: DATE‑LM is the first to bring together a large‑scale, end‑to‑end benchmark for this task on 8B‑parameter models, and to introduce homogeneous vs. heterogeneous settings with safety‑aligned distractors—addressing the design flaws of past evaluations [2] and measuring robustness in realistic scenarios.
> - Factual Attribution: Prior benchmarks in this task [3, 4] are biased toward simple lexical overlap. DATE‑LM remedies this by adding an unbiased counterfactual evaluation—systematically corrupting supporting facts —and by comparing a diverse set of attribution methods across multiple LLM backbones, yielding the first comprehensive, fair comparison in factual tracing.
>
> Together, these enhancements make DATE‑LM a comprehensive framework for developing, comparing, and improving data‑attribution techniques in LLM research. In the next version of the paper, we will be sure to include a comprehensive explanation of our designs.
>
> [1] Li et al. DataComp-LM: In search of the next generation of training sets for language models. 2024
>
> [2] Pan et al. Detecting and Filtering Unsafe Training Data via Data Attribution. 2025
>
> [3] Akyurek et al. Towards tracing knowledge in language models back to the training data. EMNLP 2022.
>
> [4] Chang et al. Scalable influence and fact tracing for large language model pretraining. 2024
>
> ---
>
> **The use of large language models (LLMs) for evaluation is not clearly explained. It is unclear whether the same LLM is used across the three application scenarios.**
>
> We thank the reviewer for highlighting the need to clarify our usage of LLMs. Below, we have provided a table of the LLMs used in our evaluation across all tasks. We will be sure to include this table in the next version of our paper.
>
> | Setting                   | Model Architecture                              | Checkpoint Source                                     |
> |---------------------------|-------------------------------------------------|-------------------------------------------------------|
> | Pre-training              | Pythia-1b                                       | Huggingface (checkpoint trained on FineWeb from [5])  |
> | Fine-tuning               | meta-llama/Llama-3.1-8B                         | Huggingface                                           |
> | Toxicity/bias Filtering   | meta-llama/Llama-3.1-8B, EleutherAI/pythia-1b   | Huggingface                                           |
> | Factual Tracing           | meta-llama/Llama-3.1-8B, EleutherAI/pythia-1b   | Huggingface                                           |
>
> [5] Yu et al. MATES: Model-Aware Data Selection for Efficient Pretraining with Data Influence Models. Neurips 2024.
>
> ---
>
> **Additionally, it is important to clarify how the accuracy of each data attribution method is ensured during the evaluation process.**
>
> We thank the reviewer for highlighting this. In order to maintain the accuracy and robustness in our evaluations, DATE-LM adopts a standardized pipeline where all methods operate on the same model checkpoints, datasets, and reference examples within each task. This uniform backbone ensures that differences in performance arise solely from the attribution methods and task designs. In particular,  DATE-LM provides:
>
> - Identical Checkpoints & Data. All methods operate on the same model weights, datasets, and hold‑out reference examples. Moreover, we release all model weights, datasets and reference examples publicly. As a result, our design is able to support robust large-scale evaluation
> - Canonical Implementations. We leverage open‑source code from Dattri [6] Library , MATES [5], and related repos, validating that each method reproduces known trends before inclusion.
> - Leaderboard & Ground Truth. DATE‑LM’s public leaderboard hosts ground‑truth labels and ranking metrics, enabling transparent comparison of attribution‑method accuracy.
> Together, this design guarantees that attribution accuracy is measured fairly and reproducibly across tasks. We will highlight these features in the next version of our paper.
>
> [6] Deng, Junwei, et al. "$\texttt {dattri} $: A Library for Efficient Data Attribution." Advances in Neural Information Processing Systems 37 (2024).

---

> > ### Comment · Reviewer_AcxT · 2025-08-07
> >
> > Thanks for your response. My concerns have been addressed.

---

### Official Review · Reviewer_x22W · 2025-07-02

**Rating:** 4
**Confidence:** 4

**Summary:**

Data attribution methods quantify the influence of training data on model outputs. This paper introduces a benchmark suite for these methods, consisting of three application scenarios: training data selection, toxicity and bias filtering, and factual attribution. The benchmark provides a unified evaluation pipeline for data attribution methods. Eight methods are assessed, including both data attribution techniques and some non-attribution baselines. The results reveal several interesting insights, notably that data attribution methods often do not outperform non-attribution baselines.

**Dataset Code Accessibility:**

Yes

**Ethical Considerations:**

No, there are no or only very minor ethics concerns

**Final Justification:**

I appreciate the author's responses, which justify the use of reference data to some extent. Since my score is already positive, I have decided not to make any further changes.

**Limitations Weaknesses:**

- The setting does not align perfectly with the objective of data attribution. Data attribution methods measure the influence of training data on model outputs. However, this paper uses these methods to assess the influence of training data on reference data, which is not generated by the model. Utilizing reference data as a proxy appears to be crucial for the benchmark's success, as it allows for automatic evaluation of attribution quality. However, this approach introduces inconsistency between evaluation results and the task goal.
- Despite the unified pipeline, the benchmark still doesn't apply to all data attribution methods (as shown in Table 1) due the the above-mentioned limitation.

**Strengths Contributions:**

- The paper is easy to follow, and the related works are properly discussed.
- The proposed benchmark effectively integrates various data attribution scenarios into a single unified pipeline, which could benefit the community by facilitating more standard evaluation.
- The paper offers interesting insights, particularly the observation that data attribution methods often cannot outperform non-attribution methods. This raises an important question regarding the conditions under which data attribution is truly beneficial. A natural extension of this work would be to further investigate and delineate the specific contexts or applications where data attribution methods provide a clear advantage.

---

> ### Author Rebuttal · Authors · 2025-07-31
>
> We thank the reviewer for their thoughtful and encouraging feedback. We are glad the reviewer found the paper to be clear and well-situated in the literature. We will address your concerns/questions below:
>
> ---
>
> **The setting does not align perfectly with the objective of data attribution. Data attribution methods measure the influence of training data on model outputs. However, this paper uses these methods to assess the influence of training data on reference data, which is not generated by the model.**
>
> We appreciate the reviewer’s observation. We agree with the reviewer that data attribution methods measure the influence of training data on model outputs. The term “outputs” is general in this context: we can measure the influence of training data on text that the model has generated (case 1) as well as probabilities for generating a text (case 2). For instance, given a text (which may or may not be generated by the model), we can measure the loss of the model on the target text, and use information to calculate training data attribution scores, as done in [1, 2, 3, 4, 5].
>
> To that end, we follow a common setup in the data attribution literature (which follows case 2): using reference data to score the influence of training examples. By using a reference set, we evaluate the influence of training data on a target data distribution, which is guided by the reference set. We subsequently use these scores to perform different downstream tasks. It is worth noting that this setup is already employed in numerous prior works, such as using data attribution towards data selection (LESS [2], TracIn[3], MATES [4]), and toxicity detection such as DABUF [5]. Finally, we note that we follow this setup since i) directly evaluating data attribution scores against a set of “ground truth” scores is often unfeasible to obtain (see [1] for additional details) and ii) our benchmark is designed with an application-driven focus (as highlighted in Section 2.2 of our paper) in order to assess the practical utility of data attribution methods. We will include the reviewer’s feedback regarding the clarity of model outputs in the next version of our paper.
>
> [1] Hammoudeh and Lowd. Training Data Influence Analysis and Estimation: A Survey. 2024.
>
> [2] Xia et al. LESS: Selecting Influential Data for Targeted Instruction Tuning. ICML 2024.
>
> [3] Pruhti et al. Estimating Training Data Influence by Tracing Gradient Descent. Neurips 2020
>
> [4] Yu et al. MATES: Model-Aware Data Selection for Efficient Pretraining with Data Influence Models. Neurips 2024.
>
> [5] Pan et al. Detecting and Filtering Unsafe Training Data via Data Attribution. 2025
>
> ---
>
> **Utilizing reference data as a proxy appears to be crucial for the benchmark's success, as it allows for automatic evaluation of attribution quality. However, this approach introduces inconsistency between evaluation results and the task goal.**
>
> We agree with the reviewer that using reference data enables automatic, reproducible, and scalable evaluation. We note that this use of reference data does not conflict with the goal of data attribution. In fact, it helps approximate real-world scenarios, where practitioners seek to identify high-quality or relevant training data based on proxy references (e.g., representative tasks or examples), which they can then apply to a downstream task, such as training data selection as noted in our framework.
> Moreover, in many cases, training on data identified via attribution methods using a reference set outperforms standard baselines that do not rely on such references—for instance, against the API-based toxicity detection baselines (see Table 4 in our paper). This demonstrates that using a reference set can aid downstream model performance, making it consistent with the task goal.

---

> > ### Comment · Reviewer_x22W · 2025-08-04
> >
> > I appreciate the author's responses, which justify the use of reference data to some extent. Since my score is already positive, I have decided not to make any further changes.

---

### Official Review · Reviewer_NWxy · 2025-07-02

**Rating:** 5
**Confidence:** 3

**Summary:**

This paper introduces DATE-LM, a unified benchmark suite for evaluating data attribution methods in Large Language Models (LLMs). The benchmark centers on three real-world LLM use cases: training data selection, toxicity/bias filtering, and factual attribution. The authors provide a modular evaluation pipeline that allows researchers to plug in different attribution techniques and test them across diverse tasks and model architectures. They also release pre-trained/fine-tuned model checkpoints to ease reproducibility and a public leaderboard to track and compare methods. Using DATE-LM, the paper conducts a large-scale empirical study of numerous attribution methods (influence-function based, gradient-based, etc.) alongside non-attribution baselines. The key finding is that no single method excels on all tasks; in fact, simpler baselines sometimes rival more complex attributors, and performance can be highly sensitive to evaluation design.

**Dataset Code Accessibility:**

Yes

**Dataset Code Comments:**

The code is accessible and clearly documented at https://github.com/DataAttributionEval/DATE-LM.

**Ethical Considerations:**

No, there are no or only very minor ethics concerns

**Final Justification:**

This work fills a clear gap by providing the first unified, LLM-centric benchmark for data attribution, which I think is a significant contribution. During rebuttal, my concerns have been addressed as well. I think the paper should be accepted.

**Limitations Weaknesses:**

- To keep experiments feasible, DATE-LM uses proxy models and datasets that approximate larger scenarios. An 8B parameter model is large, but far smaller than most cutting-edge models; similarly, fine-tuning on tens of thousands of samples is a proxy for the billions of samples in pretraining. Thus, some findings might not fully extrapolate to the largest models without caution.
- The experimental results, while extensive, are mostly reported as point estimates without confidence intervals or significance testing. The authors explicitly state that they did not include formal statistical significance analysis, focusing on directional trends. Given the computational expense, it’s understandable that multiple runs per experiment weren’t feasible. However, this leaves some uncertainty about how robust certain differences are. For example, in Table 2, the average accuracy differences between many methods are less than 1 point. Without error bars, it’s hard to tell if Rep-Sim’s 46.00 vs MATES’s 45.76 (at 10k steps) is a meaningful lead or within run-to-run noise.

**Strengths Contributions:**

- This work fills a clear gap by providing the first unified, LLM-centric benchmark for data attribution. Previous evaluations were piecemeal or lacked LLM focus, whereas DATE-LM offers a systematic and reproducible framework.
- A significant contribution is the integrated pipeline that standardizes the evaluation process. The modular design allows fair, apples-to-apples comparisons by keeping ancillary components (data, model training procedure, selection heuristic) consistent. Researchers can “seamlessly plug in” their own attribution method and evaluate it across all tasks and models with minimal overhead.
- The evaluation conducted is impressively comprehensive. The authors benchmark a broad range of methods: classical influence-function approaches, recent LLM-specific methods like LESS and MATES, gradient-based similarity methods, etc. This breadth adds credibility to their findings. Multiple LLM architectures and sizes were tested (e.g. Pythia-1B, two LLaMA 1B variants, and LLaMA-8B in the factual tracing task, as well as 1B vs 8B in other tasks).
- The paper is organized in a logical manner, and the writing is clear and accessible. The authors do a good job highlighting why each task is set up in a certain way by discussing limitations of prior approaches (this contextualization helps the reader appreciate the design choices).

---

> ### Author Rebuttal · Authors · 2025-07-31
>
> We thank the reviewer for their comprehensive review, and their acknowledgements of our framework’s contributions towards evaluation data attribution methods. We will address your concerns below:
>
> -----
> **To keep experiments feasible, DATE-LM uses proxy models and datasets that approximate larger scenarios. An 8B parameter model is large, but far smaller than most cutting-edge models; similarly, fine-tuning on tens of thousands of samples is a proxy for the billions of samples in pretraining. Thus, some findings might not fully extrapolate to the largest models without caution.**
>
> We thank the reviewer for this thoughtful comment. While we agree that 8B parameter models are smaller than frontier-scale models. However, they are also well-aligned with current practices in data attribution research. For instance, recent methods such as MATES [1], quad [2], and GREATS [4] conduct pretraining-scale experiments on models of up to 1B parameters, and instruction tuning evaluations (e.g., LESS [3]) with models up to 8B. Even in large-scale efforts such as AI2’s Large-Scale Data Selection for Instruction Tuning [5], evaluations are performed on 8B models.
>
> Given these norms, our choice of model sizes strikes a balance between feasibility and relevance in current literature. Moreover, the DATE-LM benchmark is designed to be flexible: users with greater compute resources can run our comprehensive evaluation suite on larger models by modifying the model name or path in our code. Our open leaderboard infrastructure further supports this and encourages community-driven scaling of evaluations to larger models and datasets.
>
> [1] Yu et al. MATES: Model-Aware Data Selection for Efficient Pretraining with Data Influence Models. Neurips 2024.
>
> [2] Zhang et al. Harnessing Diversity for Important Data Selection in Pretraining Large Language Models. ICLR 2025
>
> [3] Xia et al. LESS: Selecting Influential Data for Targeted Instruction Tuning. ICML 2024.
>
> [4] Wang et al. GREATS: Online Selection of High-Quality Data for LLM Training in Every Iteration. Neurips 2024.
>
> [5] Ivison et al. Large-Scale Data Selection for Instruction Tuning. 2025
>
>
> **The experimental results, while extensive, are mostly reported as point estimates without confidence intervals or significance testing. The authors explicitly state that they did not include formal statistical significance analysis, focusing on directional trends. Given the computational expense, it’s understandable that multiple runs per experiment weren’t feasible. However, this leaves some uncertainty about how robust certain differences are. For example, in Table 2, the average accuracy differences between many methods are less than 1 point. Without error bars, it’s hard to tell if Rep-Sim’s 46.00 vs MATES’s 45.76 (at 10k steps) is a meaningful lead or within run-to-run noise.**
>
>
> We thank the reviewer for understanding that multiple large-scale training runs may be infeasible due to resource constraints, a limitation shared by prior works [1, 3, 4].
>
> To ensure robust comparisons, we control randomness by using the same seed for data selection across all methods (Other than this, the results are deterministic, as we are using fixed model checkpoints).  With regards to the Table 2 results in question, we do evaluate each method using 4 different Gumbel random seeds and across 2 distinct reference datasets. As shown in Figure 2, this setup yields highly consistent trends, reinforcing the stability of our conclusions.
>
> To further assess robustness, we conducted an additional training run for less expensive methods in Table 2, each with another data selection random seed. The observed variation was minimal, less than 0.1% on average (e.g., Random: 45.34% -> 45.39%, BM25: 47.37% ->  47.26%, repsim: 45.43% -> 46.47%), indicating that differences greater than 0.2–0.3% are unlikely to be due to noise. This supports the reliability of our findings and conclusions discussed in Section 5. We will extend this analysis to all methods in the next revision.
>
> | Step   | Method  | Reference | Temp | Avg   | SciQ  | ARC-Easy | ARC-Chal | LogiQA | BoolQ | HellaSwag | PIQA  | Wino | OBQA |
> |--------|---------|-----------|------|-------|-------|----------|----------|--------|--------|------------|-------|------|------|
> | 10k    | Random  | LAMBADA   | N/A  | 45.34 | 68.9  | 43.1     | 24.4     | 27.5   | 52.0   | 40.7       | 69.0  | 53.5 | 29.0 |
> | 10k    | Random  | LAMBADA   | N/A  | 45.39 | 69.0  | 43.1     | 23.9     | 27.2   | 53.9   | 40.8       | 69.0  | 52.8 | 28.8 |
> | 10k    | BM25    | LAMBADA   | 0.5  | 45.37 | 67.6  | 43.2     | 23.5     | 26.0   | 55.6   | 40.6       | 69.0  | 53.6 | 29.2 |
> | 10k    | BM25    | LAMBADA   | 0.5  | 45.26 | 67.3  | 42.9     | 24.1     | 25.8   | 55.3   | 40.5       | 69.4  | 52.0 | 30.0 |
> | 10k    | RepSim  | LAMBADA   | 0.5  | 45.43 | 67.4  | 43.9     | 23.3     | 26.0   | 55.5   | 40.9       | 69.4  | 52.9 | 29.6 |
> | 10k    | RepSim  | LAMBADA   | 0.5  | 45.47 | 67.5  | 43.9     | 23.3     | 26.0   | 55.5   | 40.9       | 69.5  | 53.0 | 29.6 |

---

> > ### Comment · Reviewer_NWxy · 2025-08-05
> >
> > Thank you for the response. My concerns have been addressed. I think the current score is appropriate so I will keep it as it is.

---

### Note · Authors · 2025-08-12

We thank the reviewers and ACs for the engaging discussion. We highlight again that the goal of our paper is to provide a framework for conducting a large-scale evaluation of data attribution methods through important and relevant LLM application tasks. We facilitated this by benchmarking a wide range of recent data attribution methods, and by contributing a leaderboard for the community to engage with in order to promote future discussions.

We are glad that the reviewers highlight that our framework “fills a clear gap” (reviewer NWxy) in the realm of data attribution evaluation, finds our task designs well-motivated (reviewer NWxy), offers interesting findings with respect to current data attribution methods (reviewers AcxT and x22W), and finds our paper well-structured, clearly organized, and easy to follow (reviewers x22W and uP4U).

Furthermore, we are glad to have engaged in discussions to address the concerns of reviewers: such as the purpose of using a reference set (reviewer x22W), adaptability with scaling to larger models (reviewer NWxy), and justifying the evaluation of data attribution methods towards LLM application tasks and its impact in the community (reviewer uP4U). Moreover, we thank reviewer uP4U for acknowledging our rebuttal discussion and raising their score.

We hope our framework serves as a foundation for future data attribution research in LLMs, and we are glad that the reviewers’ overall sentiment towards our paper is positive.

---

### Decision · Program_Chairs · 2025-09-18

**Decision:**

Accept (poster)

**Comment:**

This paper proposes a benchmarking method for evaluating data attribution for large language models. Several reviewers endorsed this work, claiming that it fills a clear gap by providing the first unified, large language model (LLM)-centric benchmark for data attribution. Previous evaluations were piecemeal or lacked an LLM focus; DATE-LM, however, offers a systematic and reproducible framework.

I also noticed that certain reviewers raised concerns about the concept of 'data attribution'. Indeed, I believe this concept should be reconsidered in the context of LLMs. In the rebuttal, I noticed that

> …the reviewer that data attribution methods measure the influence of training data on model outputs. The term “outputs” is general in this context: we can measure the influence of training data on text that the model has generated (case 1) as well as probabilities for generating a text (case 2). For instance, given a text (which may or may not be generated by the model), we can measure the loss of the model on the target text, and use information to calculate training data attribution scores, as done in [1, 2, 3, 4, 5].

I agree that the key point is that the meaning of 'output' has shifted in the context of LLMs, which are strongly correlated with the evaluation of LLMs themselves. In supervised learning, if a model produces Y, the influential functions can be used to evaluate the relevance of the data. However, the unsupervised nature of LLMs makes the output somewhat vague. I would suggest adding clear definitions to the problem setup.


> To that end, we follow a common setup in the data attribution literature (which follows case 2): using reference data to score the influence of training examples. By using a reference set, we evaluate the influence of training data on a target data distribution, which is guided by the reference set. We subsequently use these scores to perform different downstream tasks. It is worth noting that this setup is already employed in numerous prior works, such as using data attribution towards data selection (LESS [2], TracIn[3], MATES [4]), and toxicity detection such as DABUF [5]. Finally, we note that we follow this setup since i) directly evaluating data attribution scores against a set of “ground truth” scores is often unfeasible to obtain (see [1] for additional details) and ii) our benchmark is designed with an application-driven focus (as highlighted in Section 2.2 of our paper) in order to assess the practical utility of data attribution methods.

Overall, I am satisfied with the responses. However, I am a little unsure about the following:

>  We subsequently use these scores to perform different downstream tasks.

It seems that the definition of the problem setup does not properly define the downstream tasks. I would suggest clarifying this in the revised version.

Overall thoughts: Based on the reviews, I agree that the paper is novel and significant. I acknowledge the limitations identified by certain reviewers. I think this limitation is generally not caused by the authors, but rather by the vague definition of data attribution in the LLM context. Based on the contribution, I would support acceptance, but I would suggest that the authors add certain revisions to the equations in the final version.